# MDM2-Based Proteolysis-Targeting Chimeras (PROTACs): An Innovative Drug Strategy for Cancer Treatment

**DOI:** 10.3390/ijms231911068

**Published:** 2022-09-21

**Authors:** André T. S. Vicente, Jorge A. R. Salvador

**Affiliations:** 1Laboratory of Pharmaceutical Chemistry, Faculty of Pharmacy, University of Coimbra, 3000-548 Coimbra, Portugal; 2Center for Neuroscience and Cell Biology, University of Coimbra, 3004-504 Coimbra, Portugal

**Keywords:** proteolysis-targeting chimera (PROTAC), mouse double minute 2 (MDM2), p53, cancer, targeted protein degradation, anticancer activity, ubiquitin-proteasome system (UPS), drug discovery

## Abstract

Proteolysis-targeting chimeras (PROTACs) are molecules that selectively degrade a protein of interest (POI). The incorporation of ligands that recruit mouse double minute 2 (MDM2) into PROTACs, forming the so-called MDM2-based PROTACs, has shown promise in cancer treatment due to its dual mechanism of action: a PROTAC that recruits MDM2 prevents its binding to p53, resulting not only in the degradation of POI but also in the increase of intracellular levels of the p53 suppressor, with the activation of a whole set of biological processes, such as cell cycle arrest or apoptosis. In addition, these PROTACs, in certain cases, allow for the degradation of the target, with nanomolar potency, in a rapid and sustained manner over time, with less susceptibility to the development of resistance and tolerance, without causing changes in protein expression, and with selectivity to the target, including the respective isoforms or mutations, and to the cell type, overcoming some limitations associated with the use of inhibitors for the same therapeutic target. Therefore, the aim of this review is to analyze and discuss the characteristics of MDM2-based PROTACs developed for the degradation of oncogenic proteins and to understand what potential they have as future anticancer drugs.

## 1. Introduction

The human organism, with trillions of cells, is extremely complex and fascinating [1]. Although, over the years, advances in its understanding have been made, much remains to be clarified. What is certain is that balance is crucial for its normal functions, not only internally but also within its entire environment [2]. When this is not the case, it can result in a breakdown of biological mechanisms and can cause a disruption in homeostasis, promoting the development of diseases [2].

Of all the diseases known today, cancer is one of the most concerning in recent times; this is due to its high mortality rate, with data from 2020 pointing to more than 18 million cases worldwide, causing more than 10 million deaths, making it the second leading cause of death worldwide [3]. In fact, cancer corresponds to a set of diseases characterized by the uncontrolled proliferation of any of our cells, which, by various mechanisms, evade the apoptosis process, eventually spreading and invade adjacent or even distant areas in our organism, in a process called metastasis [1,4]. More than 200 different types are currently known [5], with breast, lung, colorectal, and prostate cancers being the most commonly diagnosed [6].

Nowadays, there are multiple strategies that have been created to try to prevent the development and progression of cancer, such as radiotherapy, surgery, immunotherapy, stem cell transplant, chemotherapy, among others [7]. Focusing on chemotherapy, this is often characterized by serious adverse effects, as well as, for example, difficulty in reaching the target or the development of resistance or tolerance, given the heterogeneity and complex dynamics of the tumor environment [8].

Therefore, the search for new therapies has been incessant; recently, the targeted protein degradation (TPD), which includes the PROteolysis-TArgeting Chimeras (PROTAC), has proved to be an innovative and promising way of treating cancer, as it allows the selective degrading of proteins relevant to the development of cancer, the oncogenic proteins [9,10,11].

### 1.1. Targeted Protein Degradation

Over the past few years, huge efforts have been made to develop increasingly effective and safe cancer therapy strategies to prevent or avoid cancer development. There are several strategies available, such as the inhibition of biological targets through the use of small molecule inhibitors (SMIs) or with monoclonal antibodies (mAb) and the degradation of messenger RNA (mRNA) by RNA interference (RNAi), among others [12,13].

Recently, TPD, that is, the ability to selectively degrade target proteins, has taken on new proportions, being very promising as the future of cancer therapy; this is because degrading proteins, making use of the cellular machinery itself, bring advantages over inhibition [13]. Of the 20,000 proteins that constitute the human proteome, more than 600 are known to be related to certain types of cancer [9]. However, about 63% of these carcinogenic proteins, which include, for example, transcription factors, scaffold proteins, and membrane-bound proteins, do not have active sites or antigens for their inhibition to occur, being designated as “undruggable proteins” [9,12,14].

In this sense, the TPD made it possible to degrade these “undruggable proteins” through revolutionary therapeutic strategies, such as molecular glues or the development of heterobifunctional molecules, also known as the PROTACs [5,15].

### 1.2. PROteolysis-TArgeting Chimeras—PROTACs

PROTACs—PROteolysis-TArgeting Chimeras—are also referred to in the literature as “degraders”, “degronimids”, “PROteolysis TArgeting Peptide (PROTAP)”, and “Protein Degradation Probe (PDP)”, and came to provide an entire innovative strategy for the development of new drugs [16].

Although the initial idea of PROTACs is now already more than 20 years old, having emerged in 1999, and being concretized with the publication of the first PROTAC, for which the target was the Methionine aminopeptidase-2 (MetAP-2) in 2001 [17], it is only very recently that there has been an exponential growth in their development [18], presenting a great potential to become the future blockbuster in cancer therapy [19].

In a simplistic way, PROTACs are bifunctional molecules that allow for selective post-translational degradation of a protein of interest (POI) using the cell’s own machinery; more specifically, the ubiquitin-proteasome system (UPS) [10,20]. Its mission is to bring the POI closer to an E3 ligase, which is responsible for adding ubiquitin (Ub) units to it. In the end, this makes the cell recognize the target as something to be degraded by the 26S proteasome [20].

For the POI to be recognized by the 26S proteasome and, subsequently, degraded, it is necessary that several Ub units are covalently linked to it, forming a chain in a process called polyubiquitination [9,21].

Looking more deeply into the UPS, this is one of the most important protein degradation pathways inside the cell (present in eukaryotes) and is essential for the maintenance of homeostasis, either intracellularly or in the organism itself [21]. The discovery of the importance of this system was recognized by the Nobel Prize in Chemistry, in 2004 [19], given the role it plays in vital aspects of cell life, such as cell growth and proliferation, apoptosis, endocytosis processes and the downregulation of receptors and transporters, up to the degradation of proteins in their normal protein turnover process or for reasons of abnormality [22,23]. Thus, the UPS is primarily responsible for regulating cellular proteolysis and, consequently, for the maintenance of the transcriptome [22].

The Ub is a protein with 76 amino acid residues (aa), with an approximately globular conformation, which is highly conserved among eukaryotic species and that forms peptide bonds between one of its seven lysine residues or the amino-terminal methionine residue, with the C-terminal carboxylic acid of glycine of another Ub molecule to form polyubiquitin (poly-Ub) chains in POI [9,19,22,23].

The ubiquitination process, which consists of the post-translational modification of the substrate, requires ATP and can be divided into three steps (Figure 1) [19]:The activation of Ub by the ubiquitin-activating E1 enzyme, which forms the ubiquitin-adenylate intermediate. Subsequently, a thioester bond is generated between the active site cysteine residue of the E1 enzyme and the glycine residue at the C-terminus of Ub in an ATP-dependent process [19,21,22,24];The transfer of activated Ub from the E1 enzyme to the ubiquitin-conjugating enzyme E2 [19,22];The E3 ubiquitin-ligase enzyme catalyzes the transfer of the activated Ub from the E2 enzyme to the POI through the formation of a covalent bond [19,22].

The fate of the ubiquitinated substrate will always depend on the type of ubiquitination, i.e., the position of the mono-ubiquitination, multi-ubiquitination, or polyubiquitination, depending on the number of Ubs added, as well as the typology of the poly-Ub chain(s) formed [9,23,25,26].

Proteins with poly-Ub chains can be later recognized by the 26S proteasome complex and degraded in an ATP-dependent process [24].

The PROTACs do nothing more than direct the UPS to a certain pre-selected target and promote its degradation [10,11]. Since they are heterobifunctional molecules, giving them the designation of “chimeras”, they bind at the same time to POI and E3 ligase, giving rise to a stable ternary complex [19]. This results in a forced approximation between these two molecules, which is necessary for the E3 ligase to transfer the Ub to the intended target, being polyubiquitinated and later degraded by the proteasome, which is something that normally would not happen (Figure 2) [12,19,27].

A PROTAC can be divided into 3 parts [22]:A ligand that allows connecting to the POI;A ligand that allows binding to the E3 ligase;A spacer between the two ligands—linker.

From a drug design point of view, PROTACs are extremely versatile molecules since any of their three parts can be studied, depending on the chosen target, the typology, and the length of the linker or the chosen E3 ligase [22].

## 2. Evolutionary Perspective of PROTACs

Although the concept of the PROTACs has recently emerged, its evolution is such that we can divide it into three generations (Figure 3).

### 2.1. First Generation—Peptide-Based PROTACs (2001–2008)

The first PROTAC appeared in 2001, in a work led by Crews and Deshaies, whose target was MetAP-2 [17]. Designated as “PROTAC-1”, it covalently binds to the POI via ovalacin (angiogenesis inhibitor) and recruits one E3 ligase through the phosphopeptide, IkBα, with 10 aa [17]. However, its large size reduces the ability to get inside the cells, having to be micro-injected into them [17]. On the other hand, its phosphopeptide part makes it susceptible to intracellular phosphatases [17].

In 2003, the same group, recruiting the same E3 ligase as the previous PROTAC, published an estradiol-based PROTAC, for which the target was the estrogen receptor (ER), α isoform, and a dihydroxytestosterone (DHT)-based PROTAC, for which the target was the androgen receptor (AR) [28]. Both compounds demonstrated that they could degrade their respective targets quickly and effectively [28]. These targets are implicated in the progression of breast cancer (BC) and prostate cancer (PC), respectively [28]. Despite this, the limitations are the same as those of PROTAC-1 [28].

To overcome these obstacles, the Crews group, in 2004, created the first peptide based PROTACs able to permeate cells [29]. To this end, the new molecules, through a peptide derivative of the hypoxia-inducible factor-1α (HIF1α), recruit the Von Hippel-Lindau (VHL) E3 ligase, and presented a poly-D-arginine chain in the ligand that recruits the enzyme, allowing for an increase in cell permeability and conferring proteolytic resistance. The chosen targets were the AR and FKBP12 [29].

Over the years, new groups began to investigate, and innovative optimizations were performed, which allowed them to obtain new molecules. For example, in 2008, PROTAC-A against AR (ligand: DHT) and PROTAC-B against ER (ligand: estradiol) appeared [30]. Both PROTACs use the VHL E3 ligase, but the ligand that recruits it results from a shortening of the peptide derivative of HIF1α, becoming a pentapeptide [30]. Overcoming the difficulties in penetrating cell membranes, the two compounds were able to inhibit cell proliferation, with an increase in activity compared to previous PROTACs [30].

Overall, although this generation achieved the selective degradation of their targets, the concentrations required for this to happen were in the micromolar range [17,29,30].

### 2.2. Second Generation—Small Molecule-Based PROTACs

Given that the PROTACs, until the present date, were of peptide origin and had high molecular weights, their stability and cellular permeability were their weak points, forcing the creation of new strategies to obtain better results.

Therefore, in 2008, the Crews group presented the first and revolutionary “small molecule-based PROTAC”, which started a whole new generation of PROTACs [20].

This first small molecule-based PROTAC consisted of a non-steroidal AR ligand, and a mouse double minute 2 (MDM2) E3 ligase recruiter, nutlin-3a, connected by a linker [20]. The fact that the ligands chosen in PROTACs are small chemical molecules, substantially reducing their molecular weight, promotes their entry into cancer cells [20]. However, concentrations in the micromolar range are still required [20].

In the years that followed, new targets, ligands, and E3 ligases were chosen for the design of the new PROTACs with the aim of increasing cellular activity, reducing the concentration required for the degradation, and increasing stability, among many other aspects.

With remarkable success, small-based PROTACs have emerged to recruit other E3 ligases besides MDM2.

In 2010, the first specific and non-genetic IAP–dependent Protein ERaser (SNIPPER), a designation given to PROTACs that recruit the cellular inhibitor of apoptosis protein 1 (IAP) E3 ligase–emerged [31,32]. To recruit this enzyme, small ligands, such as bestatin (MeBS) [31], MV1 [33], and e LCL161 derivatives [34], were used.

In 2012, the incorporation of small molecular VHL ligands, such as 3-fluoro-4-hydroxyprolines, into PROTACs resulted in a reduction in their molecular weight compared to the previous generation of VHL-based PROTACs [35].

In 2015, after the discovery of the link between immunomodulatory drugs, such as thalidomide, pomalidomide, and lenalidomide, with cereblon (CRBN) E3 ligase, allowed for the creation of the CRBN-based PROTACs [36]. Currently, this is the largest existing group of PROTACs, given the excellent results obtained in the degradation of several targets in cancer [9]. It is noteworthy that, in 2019, PROTAC ARV-110 (AR), designated as “Bavdegalutamide”, and ARV-471 (ER), created by Arvinas, Inc. for the treatment of PC and BC, respectively, are the first PROTACs to enter into clinical trials and both are currently in phase II [37,38]. More recently, PROTAC ARV-766 (AR) has entered phase I clinical trials, and, similarly to ARV-471, it is administered orally [39].

Over the last few years, new PROTACs have emerged, presenting increasingly better features in terms of allowing oral administration, good cell permeability and solubility, as well as a long duration of action and efficacy at nanomolar concentrations, making them an excellent alternative to the disadvantages presented by conventional therapies, such as SMIs or mAbs [10,11,24].

### 2.3. Third Generation—Spatiotemporal Controllable PROTACs

This new generation of PROTACs results from the application of modern technologies to obtain an improvement in specificity, minimizing the possible cytotoxic and off-target effects as much as possible, ensuring greater safety associated with the treatment using these molecules, and an increase in efficacy and duration of action, or to improve their pharmacokinetic aspects [10,11,24].

One of the first examples is the phosphoPROTAC, created by the Crews group in 2013, in which a peptide sequence is incorporated, which is phosphorylated only by a particular type of tyrosine kinase, allowing for its activation, when phosphorylated, only in cells that contain this enzyme, with a consequent increase in specificity [40].

With the objective of obtaining a controlled protein degradation, opto-PROTACs appeared; these have a photolabile caging group, nitroveratryloxycarbonyl, which is a group that blocks the action of the PROTAC [41]. However, when the molecule undergoes irradiation by a certain type of radiation, it loses this labile group, activating the PROTAC and promoting the degradation of the POI [41].

With a slightly similar mechanism, the PHOTACs emerged, which, when irradiated by a certain type of radiation, change their conformation, going from a state of inactivation to activation [42] or vice-versa [43].

Both technologies make it possible to have temporal and spatial control over the action of the PROTACs, thus, ensuring that they act in the right place, at the right time, and in the right concentration, increasing their safety and efficacy [42,43].

The so-called in-cell click-formed proteolysis-targeting chimeras (CLIPTACs) allow for better bioavailability because this strategy consists of dividing the PROTAC into two units which are joined at the site of action, promoting the ubiquitination of the POI [44].

More recently, new ideas have emerged, such as conjugating antibodies with PROTACs, and forming the antibody-PROTAC conjugate (APC) for the treatment of cancer in order to allow for the delivery of the drug to the cancer cells in a specific way [45].

**Figure 3 ijms-23-11068-f003:**
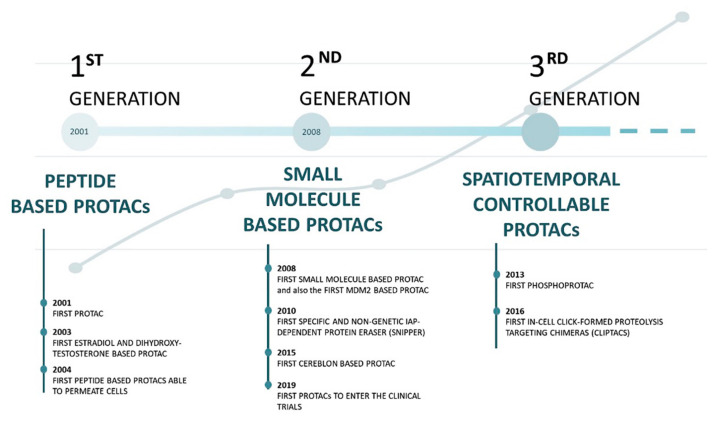
Timeline of the most relevant proteolysis-targeting chimeras (PROTACs) discoveries, and their exponential growth in recent years [10,17,20,28,30,31,36,37,38,40,44].

## 3. Discovery of PROTACs

The discovery of new PROTACs is intricately linked with the choice of the target and the E3 ligase since this will define the ligands to be linked together through a linker in the PROTAC.

### 3.1. Targets

The number of targets for the PROTACs, mainly in the cancer area, over the last few years, has increased dramatically; in general, most targets include overexpressed oncogenic proteins [12], for example, nuclear receptors (ER [28], AR [28], retinoic acid receptor alpha (RARα) [46]), protein kinases (protein kinase B (PKB) [47], BCR-abl [48], bruton’s tyrosine kinase (BTK) [49], cyclin dependent kinase 6 (CDK6) [50]), transcriptional regulators (bromodomain containing 4 (BRD4) [51]), and cellular metabolic enzymes (MetAP-2 [17]), just to name a few among many others.

### 3.2. E3 Ligase

Currently, it is known that the human genome contains approximately 8 genes for encoding the E1 enzyme [52], 50 genes for the E2 enzyme [25,52], more than 90 genes for the deubiquitinating enzymes [25,52], and over 600 genes for encoding the ubiquitin-ligase E3 enzyme [9,12,25,52].

Of all the existing E3 ligases to date, no more than 10 have been reported to be recruited by the PROTACs [9], revealing their great potential for future development. The most explored E3 ligases nowadays are CRBN; VHL; IAP; and MDM2 [12].

Aspects, such as differences in structure or function, allow for the classification of the large E3 ligase family into several subtypes: the HECT (homologous to the E6AP carboxyl terminus) type [9,53]; RING-Finger (really interesting new gene) type [9,53]; U-box type [52,53]; and the RBR (RING1-in-between-RING2) type [9,53].

Among all E3 ligase subgroups, the RING-type has the majority of members [53] (~500 enzymes) [9] and is characterized by the direct transfer of Ub from the E2 enzyme, which is linked to the E3 ligase, to the POI (unlike the HECT type, in which an intermediate (Ub-E3 ligase) is formed prior to transfer) [9,53,54]. The RING-type E3 ligases can be classified into monomeric, which have the ability to self-ubiquitinate, which is a group that includes the MDM2 E3 ligase, and multi-subunits [9,53].

Taking this into account, this review aims to explore and analyze all existing MDM2-based PROTACs, that is, PROTACs that recruit MDM2 since this enzyme is known to be involved in the regulation of p53 levels, an important tumor suppressor, and, in this sense, PROTACs, by recruiting MDM2, also inhibit it, which may bring some additional advantages over other types of PROTACs [12].

## 4. PROTACs That Recruit MDM2 E3 Ligase

Although presenting very promising results, the first generation of PROTACs, regarding the degradation of key proteins in tumor development [17,29], has some disadvantages, mainly due to high molecular weight, which ends up compromising its stability and cellular permeability, such as hindering the entire process of synthesis and purification [20]. These aspects are extremely relevant when the aim is to develop a drug since a complex synthesis and purification process causes disadvantages for the pharmaceutical industry and, thus, compromises its entry into the market [55].

The creation of smaller PROTACs, using small chemical ligands, as well as linkers of a chemical origin, is a way to circumvent the problems presented by the previous generation, as was seen with the creation of the first small-based PROTAC in 2008, which, interestingly, was also the first MDM2-based PROTAC [20].

When this first MDM2-based PROTAC was created, the main advantage revealed at the time was that it only allowed for a reduction in its dimensions [20]. However, recruiting the MDM2 E3 ligase brings more benefits in terms of cancer treatment since it is responsible for the regulation of the tumor suppressor p53, which has a relevant role in the development of a high number of cancers [7,25].

### 4.1. MDM2/p53 Cycle and Cancer

Over the life of a cell, the interactions it has with the environment can generate stressful conditions, such as oxidative stress or even damage to its genetic material [56]. To prevent these situations from becoming harmful to the organism, such as the development of tumor cells, there is an increase in intracellular levels of p53 [7]. This increase leads to the activation of signaling cascades that can culminate in a series of biological responses, such as cell-cycle arrest, DNA repair, cell senescence, or apoptosis [7]. In addition, p53 also plays an active role in regulating metabolic processes and autophagy processes [57].

By being able to trigger and regulate any of these biological responses, p53 is extremely important in suppressing the development of cancer cells (Table 1) [58].

The p53 protein, for which the name is derived from its molecular weight of 53 kDa, containing a total of 393 aa [7], is also known as the “guardian of the genome” [58] and was discovered in 1979 [59].

The mouse double minute 2 (MDM2) enzyme is a RING-Finger-type E3 ligase, with 90 kDa, for which its natural substrate is p53 [60]. MDM2 has a total of 491 aa and can be divided into 6 domains (Table 2) [58].

Under normal conditions, intracellular levels of p53 are reduced due to the presence of MDM2, an ubiquitin ligase that functions as a negative regulator of the tumor suppressor [58]. In this case, MDM2 interacts with p53, preventing it from binding to DNA and activating the transcription of several tumor suppressor genes [7].

In a simplistic way, the MDM2-p53 interaction involves three points [61]. The first is the interaction of the MDM2 N-terminal transactivation domain, which results in the inhibition of p53 binding to transcriptional machinery [61]. The second interaction is established between the MDM2 acidic domain and the specific DNA-binding domain of p53, which is essential for suppressor ubiquitination [61]. The third interaction is made between the MDM2 N-terminus and the p53 C-terminus, which contains lysines that are ubiquitinated by the MDM2 RING domain [61].

As a result of this interaction, p53 polyubiquitination occurs, followed by its degradation by the 26S proteasome [7]. Thus, p53 levels are kept constant, and its antitumor activity is inhibited [7].

However, stressful situations, such as hypoxia, DNA damage, osmotic shock, among others, prevent the binding between p53 and MDM2 through the oligomerization of the suppressor [7]. This disruption results in the transport and accumulation of p53 in the cell nucleus, with several chemical modifications to make it not only more stable, and thus increase its half-life time, but also increase its DNA binding capacity, activating gene transcription, like p21, which inhibits the cyclin-dependent kinase complexes (CDK), which leads to cell-cycle arrest (in the G1, G2, and S phases of the cell cycle), allowing the cell to repair the damage [7]. In more severe cases, pro-apoptotic factors can be activated, culminating in the release of cytochrome C and other molecules into the cytoplasm, and thus causing programmed cell death, also known as apoptosis [7].

Among the genes activated by p53 when it is present in the nucleus, one of them is the MDM2 gene itself, allowing for the increase in cellular levels of ligase to restore normal levels of the tumor suppressor through a negative feedback mechanism [58]. By polyubi-quitinating p53 and promoting its degradation by UPS, MDM2 is responsible for its regulation, thus, controlling the intracellular levels of this important protein [7]. In this sense, any deregulation of the MDM2/p53 cycle can promote or suppress the development of tumor cells [7].

Cancer cells have several mechanisms to deactivate p53. The most common mechanisms are the overexpression of MDM2 or mutations in the p53 gene (TP53, located on chromosome 17) [7,8]. The overexpression of MDM2 results in an increase in the rate of p53 degradation, and, consequently, a reduction in its levels, preventing it from performing its antitumor functions and increasing the risk of tumor development [7], as confirmed in a study in which there was a 100% incidence of tumorigenesis in MDM2-transgenic mice containing multiple copies of the MDM2 transgene inserted into a single site within the genome [62].

Nowadays, it is known that, in several types of cancer, there is an overexpression of this E3 ligase; of note, in urogenital carcinomas, breast and brain cancers, and sarcomas, among other cancer types, hence it is considered an oncogenic E3 ligase [7]. As for p53 mutations, approximately 50% of all human cancer have some type of change in the gene of this suppressor [8].

With the aim of combating the development of tumors, some therapeutic strategies have been developed to restore normal levels of p53 and MDM2 within cells [53].

At the moment, several MDM2 inhibitors are already known, such as nutlins [63], cis-imidazole compounds (which have already been shown in clinical trials to increase intracellular p53 levels and inhibit the development of cancer cells, being promising in the treatment of lung and colon cancers), multiple myeloma (MM), and others [58].

In addition to nutlins, there are also other compounds capable of inhibiting the MDM2; however, all of them end up presenting some of the limitation characteristics of the SMIs, such as the development of resistance, tolerance, toxicity, an inability to inhibit targets without an active site or mutated proteins, etc. [53].

With the discovery and development of PROTACs, this new drug discovery strategy could revolutionize the treatment of cancer, given the numerous advantages they present over conventional therapies.

Although the number of PROTACs recruiting MDM2 is still reduced, they have enormous potential in certain types of cancer, with a focus on those in which both the POI and E3 ligase are overexpressed [64]. This is because an MDM2-based PROTAC, in addition to having the ability to promote POI degradation by recruiting MDM2 with this binding ends up preventing its main function, that is, it binds to p53, leads to an increase in intracellular levels of the suppressor (Figure 4) [12].

Therefore, with MDM2-based PROTACs, it is possible not only to degrade the target oncogenic proteins, but also prevents MDM2 from binding to p53, activating its tumor suppressor action [12]. This duality of effect presented by this type of PROTAC makes their use in cancer therapy of great interest; they may become a great therapeutic weapon in the coming years, and in this sense, from now on, they will be the object of study of the present review.

### 4.2. MDM2-Based PROTACs—Classification

Although there is no classification for MDM2-based PROTACs, they can be classified from several viewpoints (Table 3).

Therefore, we can group them according to:The target they degrade, for example, if it is a nuclear receptor or a transcription factor [18];The type of cancer they want to treat [18];Their origin, that is, if they are peptides with aa chains in their composition or if they use chemically-synthesized molecules, or a mixture of both, for example [18];The nature of the linker used to join the two ligands, which can be, among other factors, a polyethylene glycol (PEG) based linker or with aliphatic chains to establish the link [18];Their mechanism of action, considering whether they degrade a target different than the chosen E3 ligase or if, for example, the target is the E3 ligase itself (HOMO PROTACs) [65];Other features of MDM2-based PROTACS.

**Table 3 ijms-23-11068-t003:** MDM2-based PROTACs. Summary table of all MDM2-based PROTACs created to date for the treatment of several types of cancer.

Name	Year	Cancer Type	POI	POI Ligand	Linker	MDM2 Ligand	Results	[Ref]
**MDM2-BASED PROTACs**								
**A**	2008	PC	AR	Selective Androgen Receptor Modulator (SARM)—hydroxyflutamide	PEG-based linker	Nutlin-3	↓AR Micromolar Potency	[20]
**B**	2018	NHL	BTK	Inhibitor Ibrutinib/spebrutinib	Distinct types	Nutlin-3 or RG-7112	No significant degradation	[66]
**C**	2019	Hematologic and solid tumors	BRD4	Inhibitor JQ1	PEG-based linker	Idasanutlin	↓BRD4 (Dmax = ~98%) Nanomolar Potency: (Cmax= 100 nmol/L) ↓c-Myc (~85%) ↑p53/p21	[67]
**D**	2019	BC	PARP1	Inhibitor Niraparib	PEG-based linkers	Nutlin-3	↓PARP1(52%,24 h) Inhib. cell growth (80–90%,48 h) Cellular selectivity	[68]
**E**	2019	BC	TrkC	Inhibitor Dasatinib	PEG-based linkers	Nutlin-3a	No Significant degradation	[69]
**F**	2019	BC	ERRα	Inhibitor XCT790	Alkyl chains	Nutlin-3b	No significant degradation	[70]
**G**	2019	BC Lymphoma MM	CDK6	Inhibitor Palbociclib	PEG-based linkers	Nutlin-3b	No significant degradation	[71]
**H**	2022	BC	HSP 90	Inhibitor BIIB021	Distinct types	Idasanutlin	No significant Cell growth inhibition	[72]
**I1**	2020	NSCLC	EGFR mutant	Inhibitor XTF-262	Alkyl chain (No carbonyl, 12C)	Idasanutlin	No Degradation (DC_50_ > 2000 nM)	[73]
**I2**					Alkyl chain (12C)		↓EGFR^MUT^ Nanomolar potency: (H1975 cells: DC_50_ = 264 nM) Target Selectivity	[73]
**I3**					Alkyl chain (10C)		↓EGFR^MUT^ Nanomolar potency: (H1975 cells: DC_50_ = 77 nM) Target Selectivity	[73]
**I4**					Alkyl chain (8C)		No Degradation (DC_50_ > 2000 nM)	[73]
**I5**					Alkyl chain (6C)		No Degradation (DC_50_ > 2000 nM)	[73]
**HOMO-MDM2-BASED PROTACs**								
**J**	2021	NSCLC	MDM2	Derivative of Nutlin-3	Distinct types	Derivative ofNutlin-3	**11a:** + potency ↓MDM2 (>95%, 24h) ↑p53 (DC_50_ = 1.0 µM) **Enantiomer 11a-1:** potent in vivo antitumor activity	[65]
**PEPTIDIC-MDM2-BASED PROTACs**								
** ^PMI^ ** **BCR/Abl-R6**	2022	Philadelphia chromosome-positive (Ph+) leukemia	Bcr/Abl	N-terminal helical region of the tetra-merizetion domain of Bcr/Abl	-	PMI—peptide inhibitor of the p53-MDM2 Interaction	↓BCR/Abl ↑p53 OBS: C-terminal Arg-repeating hexapeptide	[74,75]

LEGEND: AR—androgen receptor; Arg—arginine; BC—Breast Cancer; BRD4—Bromodomain-containing protein 4; BTK—Bruton’s tyrosine kinase; CDK6—Cyclin-dependent kinase 6; EGFR—Epidermal growth factor receptor; ERRα—Estrogen-related receptor α; HSP-90—heat shock protein 90; MDM2—Mouse doble minute 2; MM—Multiple myeloma; Mut—mutant; NHL—Non-Hodgkin’s lymphoma; NSCLC—Non-small cell lung cancer; PARP1—Poly (ADP-ribose) polymerase-I; PC—prostate cancer; PEG—polyethylene glycol; TrkC—Tropomyosin receptor kinase C.

### 4.3. The First MDM2-Based PROTAC

The first MDM2-based PROTAC, which was also the first small-based PROTAC ever created (PROTAC A), recruits the E3 ligase through the presence (at one end) of a small chemical ligand, nutlin, an inhibitor of MDM2 [20]. At the opposite end, it has a selective AR modulator (SARM), hydroxyflutamide (Ki = 4 nM), and both parts are linked together through a linker of chemical origin, which is a soluble derivative of PEG, thus, achieving a PROTAC with smaller dimensions than the previous generation (Figure 5) [20].

The choice of a small chemical ligand capable of recruiting an E3 ligase is essential to reduce the size of PROTAC, and this was possible through the incorporation of nutlin-3 [20].

Nutlins, a group of small MDM2 inhibitors based on a cis-imidazole center, appeared in 2004 and consist of nutlin-1, nutlin-2, and nutlin-3, for which the mechanism of action is the inhibition of the formation of the MDM2/p53 complex, leading to increased levels of the tumor suppressor (Figure 6) [63]. This is due to its structure, with the halogenated phenyl groups mimicking the hydrophobic aa residues of p53 [63]; more specifically, the two halogenated phenyl rings replace the leucine 26 e tryptophan 23 and the isoproproxide group, or the ethyl ether group: phenylalanine 19 of p53 [58].

Among the three, nutlins-3 is the most specific and, therefore, the one with the most studies in the area of cancer [58]. This consists of a racemic mixture in which nutlin-3A, the most potent enantiomer, is 150 times more potent than the other enantiomer, nutlin-3B [63]. This stereoselectivity is due to the ability of nutlin-3A to get closer to the MDM2 binding site, improving the binding [58].

Thus, starting from a triaryl imidazole substrate, it was possible to obtain a racemic mixture of nutlin-3, which was subsequently reacted with PEG, previously linked to SARM, obtaining, in the end, a diastereomeric mixture of the so-called “SARM-nutlin PROTAC” [20].

With this composition, 10 µM of the present PROTAC was demonstrated in HeLa cells—human cervical carcinoma cells transiently expressing the androgen receptor—to be able to permeate cell membranes, binding simultaneously to MDM2 and to the target, AR, promoting the ubiquitination of the latter, and, consequently, its intracellular degradation, verifying a reduction in the levels of this target protein [20], which is known to promote the growth of tumor cells in the prostate [76].

Although the ability to induce AR degradation has only been demonstrated from a qualitative viewpoint, proving this degradation to be proteasome-dependent [20], studies to determine the impact on p53 levels, or to determine the degradation constant at 50% (DC_50_) (that is, the concentration necessary to observe a degradation of 50% of target protein, relative to control levels [65]), were not performed. Nevertheless, this PROTAC A was extremely important as it opened the door to a new group of PROTACs.

However, despite the satisfactory results obtained, and the demonstration of the possible potential of the use of MDM2, the development of MDM2-based PROTACs was not as pronounced as that of others; this is because of the emergence of small ligands, which boosted the development of PROTACs to recruit the VHL, IAP, and CRBN E3 ligases [67].

### 4.4. MDM2-Based PROTAC—Hematological and Colorectal Cancers

Even so, 10 years after the first MDM2-based PROTAC, there was a publication of the study carried out with several PROTACs whose target was Bruton’s tyrosine kinase (BTK), with some of them recruiting MDM2 through nutlin-3 or the RG-7112 inhibitor (Figure 7) [66].

All cells of the hematopoietic system, except T lymphocytes, have the BTK enzyme, which participates in the development, differentiation and signaling processes of B lymphocytes [66]. This enzyme is responsible for promoting the development and survival of leukemic cells and their interactions with other cells, participating in multiple signaling pathways, such as the B cell receptor (BCR) [77]. The inhibition of BTK, using the inhibitor ibrutinib, is a form of treatment for non-Hodgkin’s lymphoma (NHL) [66]. Unfortunately, several cases of resistance prevent it from being used in the treatment of various lymphomas [66].

To overcome this disadvantage, the development of a PROTAC capable of degrading BTK by recruiting it through ibrutinib or spebrutinib could be useful in the treatment of NHL.

Based on the concentrations needed to obtain 50% of maximum cell proliferation inhibition (GI_50_), tests on the same cell line have shown that recruiting MDM2 (PROTAC B3: GI_50_ = 138.7 nM) is not as effective as recruiting the CRBN E3 ligase, (“P131”: GI_50_ = 1.5 nM), or even the inhibitor itself (Ibrutinib: GI_50_ = 2.5 nM) [66].

This proves that, depending on the contexts of structure, ligands, target, cancer type, or cell, the choice of the E3 ligase for recruitment is not indifferent [66].

In 2019, as a result of the work carried out by the Crews group, the first PROTAC capable not only of degrading the target but also of stabilizing p53 in concentrations on the nanomolar scale (and in a synergistic way) appeared, proving the great potential of MDM2-based PROTACs as an anticancer therapy [67].

Designated by the authors as “A1874”, PROTAC C degrades bromodomain containing 4 (BRD4), a protein known to be involved in the development of hematological and solid tumors, which has already been studied as a target with PROTACs that bind to other E3 ligases (CRBN and VHL) [67].

To be able to recruit BRD4, at one of the ends of A1874 resides JQ1, an inhibitor of this protein, which, in turn, is linked through a PEG-based linker with 13 atoms to idasanutlin, a known inhibitor of MDM2 responsible for recruiting it and promoting target degradation, and simultaneously stabilizing p53 levels (Figure 8) [67].

The BRD4, considered an oncogenic protein, belongs to the bromodomain and extra-terminal domain (BET) protein family and is crucial in several biological processes, such as mitosis since it is necessary for the formation of the chromatin structure in the daughter cells [78]. In addition, BRD4 recruits essential proteins to control the expression of those genes responsible for cell proliferation that are especially relevant in the M and G1 phases of the cell cycle, as c-Myc [79,80].

In order to inhibit the proliferation of tumor cells, in 2010, JQ1 was reported [80]. JQ1 is one of the first compounds capable of inhibiting, with nanomolar potency, the binding of BRD4 with DNA, consequently preventing the development of cancer cells (to which it activated the process of apoptosis, being, currently, a therapeutic weapon in acute leukemia [80]).

In 2013, the RG7388 inhibitor, more commonly known as idasanutlin, was discovered [81], and was a second generation of the nutlin family capable of inhibiting the binding between MDM2 and p53 and resulted from the study of the different stereochemical configurations of the pyrrolidine group, obtaining a compound that is much more potent and selective than the previous ones [81].

Idasanutlin has already performed several clinical trials [82], in which it has been shown to be effective, with its most recent phase III study (NCT02545283) completed in 2020, having been tested in association with cytarabine in the treatment of acute myeloid leukemia [83].

Thus, by linking the methoxyphenyl group of idasanutlin with the diazepine ring of the JQ1 inhibitor through the PEG linker, forming PROTAC C [67], it was possible to reveal a whole set of aspects that demonstrate the enormous potential of MDM2-based PROTACs in relation to the use of inhibitors in therapeutics of cancer:The reduction of target protein levels without inducing protein up-regulation and impacting the signaling pathway. The first cell line used was the HCT116 colon cancer cells, given the importance of the BRD4/c-Myc signaling pathway in this type of tumor [67]. When subjected to increasing concentrations of PROTAC C over a 24 h period, it was possible to observe a reduction in the BRD4 levels in a dose-dependent way, with a maximum of around 100 nmol/L, which corresponds to a maximum degradation of 98% relative to the control levels, as well as a DC_50_ value of 32 nmol/L, lower than that obtained by previous MDM2-based PROTACs [67]. Under equivalent conditions, treatment with the JQ1 inhibitor alone did not lead to a decrease in BRD4 levels [67]. This could be attributed to the fact that high concentrations of JQ1 induce up-regulation of the target protein [67]. This was not detected with higher concentrations of PROTAC C, which is also an advantage even in relation to other types of PROTAC, in which the hook effect* is verified [67]. Furthermore, the present PROTAC has been shown to reduce the expression of c-Myc (85%), which is directly related to BRD4 levels; in a way, this is superior to the results achieved with JQ1 (70%) without genetically modifying the cell [67];

* Hook effect: For PROTAC to be able to exert an effect, it needs to bind with two different ligands, forming a stable ternary complex. However, when using high concentrations, the formation of binary complexes (PROTAC-TARGET or PROTAC-E3 LIGASE) is favored, reducing the therapeutic effect [19].
Synergistic Effect—increase in p53 suppressor levels, with the inhibition of cell development. The treatment of HCT116 with PROTAC C demonstrated (in addition to degrading BRD4 and decreasing c-Myc expression) that it is capable of increasing p53 values by 5.9 times compared to the control level, activating the subsequent signaling pathway and resulting in reduced cell viability by about 97% [67]. Compared with the use of idasanutlin, the increase in p53 levels with this inhibitor was slightly higher but with no effect on the target protein, reducing cell viability by 62% [67]. With JQ1, p53 levels remained unchanged; yet, since it reduces the c-Myc pathway, it ends up achieving a 25% loss of cell viability [67]. In summary, only PROTAC C is able to combine the actions of the two inhibitors in a way that is superior to the sum of their isolated effects, allowing it to be used with a single compound to obtain an anti-proliferative synergism with nanomolar concentrations (>100 nmol/L) [67];The synergistic effect is dependent on cellular context. The same characteristics above were confirmed when melanoma cell lines (A375) were treated with PROTAC C, which caused a 98% decrease in cell viability, a value that is also greater than the sum of the isolated effects of JQ1 (15%), and idasanutlin (64%), showing a synergistic effect [67]. However, both cell lines studied have wild-type p53. When PROTAC C was evaluated in cell lines with mutations in the p53 gene (Daudi cells: hematological cancer cells; colon cancer cells: HT-29; p53-/- HCT116), it showed a decrease in its ability to reduce cell viability, which was going to be in the order of 20–30% [67]. In this type of cell, VHL-based PROTAC proved to be the best option [67].More difficulties in developing drug resistance. Given that PROTAC presents a bifunctional interaction with two distinct mechanisms of action (on the one hand, the degradation of BRD4 with the reduction of cell proliferation, on the other hand, the increase of the tumor suppressor p53), even if a mutation occurs in one of the proteins at which PROTAC binds, it ends up not compromising its entire therapeutic activity, which is an advantage relative to the inhibitors, in which all its activity is annulled [67].

Colorectal cancer (CRC) is one of the types of cancer in which the BRD4 protein is overexpressed [78]. According to the World Health Organization, CRC is the third most common type of cancer worldwide and is responsible for the deaths of about one million people a year [84].

Therefore, based on the results presented above, several studies were conducted to assess the therapeutic effect of PROTAC C on CRC. These studies demonstrated that the compound is cytotoxic to both primary CRC tumor lines, pCan1, and HCT116 cells [78]. When cells are subjected to a treatment with 100 nM of PROTAC after 48 h of exposure, there is a delay in the cell cycle, increasing the population in the G1 phase as well as creating a reduction in cell proliferation and viability [78]. It was also found that with this concentration, it is possible to compromise cell migration and invasion, in addition to having verified the activation of the apoptosis process, resulting in the death of the cancer cells [78]. It should be noted that all these anticancer activities were more potent than those achieved with conventional inhibitors, such as JQ1 [78]. Such activity results from the degradation of BRD4 without affecting the expression of its mRNA and is associated with the elevation of p53 levels as well as the triggering of oxidative damage in the cell [78]. In addition, PROTAC was able, in vivo, when administered orally to immunocompromised mice, to significantly reduce colon cancer xenograft growth [78].

In this way, we are facing the first PROTAC that proves the enormous potential that MDM2 recruitment can offer, overlapping, in many aspects, with conventional SMIs.

### 4.5. MDM2-Based PROTAC—Breast Cancers

With the aim of treating BC, several targets were chosen for the development of the MDM2-based PROTACs, as follows.

#### 4.5.1. Poly (ADP-Ribose) Polymerase-1 (PARP1)

In order to promote the apoptosis of cancer cells, Yu Rao’s research group created an MDM2-based PROTAC that targets PARP1 to obtain a new therapeutic weapon in BC [68].

Therapies targeting PARP1 have been used for the treatment of ovarian cancer and BC [85]. However, triple-negative breast cancer (TNBC) is a heterogeneous type of cancer with a poor prognosis and is resistant to existing therapies; therefore, new forms of treatment are urgently needed [68].

The PARP1 protein, an adenosine diphosphate (ADP)-ribosyltransferase, is a member of the poly (ADP-ribose) polymerases (PARPs) enzyme superfamily, which is required for DNA damage repair [85]. In a double-strand break situation, PARP1 can efficiently recognize and bind to the damaged site and promote its repair through a process of base excision [68]. In this situation, PARP1 transfers negatively charged ADP-ribose units to several other nicotinamide-adenine dinucleotide (NAD^+^) proteins [85]. This causes DNA repair proteins to be recruited to the site [85].

The use of molecules, such as olaparib, niparib, or iniparib, inhibit PARP1 preventing the repair of possible breaks in the DNA double strand, affecting the vitality and replication of the cancer cells [85].

Unfortunately, there are several mechanisms by which cancer cells can develop resistance, attenuating the effect of PARP1 inhibition [85]. Therefore, PROTAC D was created, which results from the combination of part of the niparib inhibitor with nutlin-3 through a PEG chain (Figure 9) [68].

When tested on TNBC cell lines, like MDA-MB-231, MDM2-based PROTAC D demonstrated PARP1 degradation, causing a 52% decrease over the control cell levels [68].

After incubating MDA-MB-231 cells with 10 µM of PROTAC D for a period of 24 h, cell growth was inhibited by about 70%, increasing to 80–90% after 48 h [68]. In comparison, after 48 h, none of the inhibitors alone could overcome the PROTAC results, as both niparib and nutlin-3 showed inhibitions in the order of 30% [68]. Even compared to other inhibitors, such as olaparib or veliparib, for the same cell line, PROTAC D proved to be five times more potent [68].

Knowing that PARP1 degradation can result in cellular apoptosis, flow cytometry analysis showed that cells effectively activated their apoptotic pathways when in the presence of 10 µM for 24 h, culminating in the cell death of MDA-MB-231 cells [68]. Another advantageous aspect of the present PROTAC is that when tested in other cell lines, as in normal breast epithelial cells, there was no change in PARP1 levels, causing no cytotoxic effects, which demonstrates that PROTAC C has a high selectivity, something that is often not seen with conventional inhibitors, compromising the safety of the treatment [68].

Based on this, the present compound has immense potential to become a revolutionary treatment in TNBC.

Apart from PROTAC D, although without remarkable results, other PROTACs were developed for the treatment of BC in an attempt to promote the degradation of several key carcinogenic targets (Figure 10).

#### 4.5.2. Tropomyosin Receptor Kinase C (TrkC)

Created in 2019, PROTAC E1 and E2 recruit TrkC, a protein with high expression in several types of metastatic cancer, such as BC, melanoma, glioblastoma, etc., since its activation promotes metastasis and cell growth [69]. The present PROTACs recruit TrkC through its inhibitor, dasatinib, which was approved in 2006 for the treatment of Philadelphia chromosome positive chronic myeloid leukemia [69]. To recruit MDM2, nutlin-3a was chosen [69]. Both inhibitors are linked together through a 13-atom based PEG linker in PROTAC E1 and 19 atoms in PROTAC E2 [69]. When used in cell lines that overexpress TrkC, only PROTAC E2 is shown to degrade the target [69]. In cell lines with normal expression, none of them demonstrated the degradation of TrkC, making it less effective than the CRBN-based PROTAC for the same target [69].

#### 4.5.3. Estrogen-Related Receptor α (ERRα):

The estrogen-related receptors are a subgroup of the large family of nuclear receptors, with ERRα being the first to be discovered and being the most homologous to Erα [70]. As a result of the multiple interactions that ERRα establishes with other proteins, it is responsible for the regulation of metabolic homeostasis, and has been identified as one of the factors that allows cancer cells to adapt metabolically, allowing their proliferation, metastasis, and resistance to therapies [70]. It was recognized that in TNBC and Her2+ BC, the ERRα is overexpressed, and its inhibition prevented tumor progression [70]. The PROTACs F1 and F2 were designed to degrade ERRα. To recruit it, an analog of the inhibitor XCT790, named “compound 1”, linked to nutlin-3 through an alkyl chain of two carbons (F1) and five carbons (F2), was used [70]. However, the degradation of the ERRα generated was not significant [70].

#### 4.5.4. Cyclin-Dependent Kinase 6 (CDK6)

The CDK6 is a protein with significant importance at the cell-cycle level, and it is found in large amounts in certain types of cancer, like BC, resulting in a resistance to inhibitors [71]. To promote its degradation as an alternative to inhibition, three MDM2-based PROTACs, the G1, G2, and G3, were synthesized [71], recruiting the CDK6 through the inhibitor palbociclib, linked to nutlin-3b by a PEG-based linker of various sizes [71]. These three PROTACS failed in their mission to degrade CDK6 [71].

#### 4.5.5. Heat Shock Protein 90 (HSP90)

There are several types of cancer in which the HSP90 protein is excessively activated, and as this is a protein responsible for regulating, stabilizing, and activating many other proteins, some of which are even considered oncogenic, like epidermal growth factor receptor (EGFR) or human epidermal growth factor receptor 2 (HER2), any deregulation in the signaling pathways of the cells can lead to cancer progression [72]. Therefore, to carry out the degradation of this protein based on the fact that its inhibition has a moderate anticancer effect, although with some toxic effects, several MDM2-based PROTACs (PROTACs H) were developed with different types of linkers, which recruit the target protein through the BIIB021 inhibitor and MDM2 through idasanutlin [72]. However, PROTACs that recruit E3 ligase CRBN are more effective in degrading HSP90 [72].

**Figure 10 ijms-23-11068-f010:**
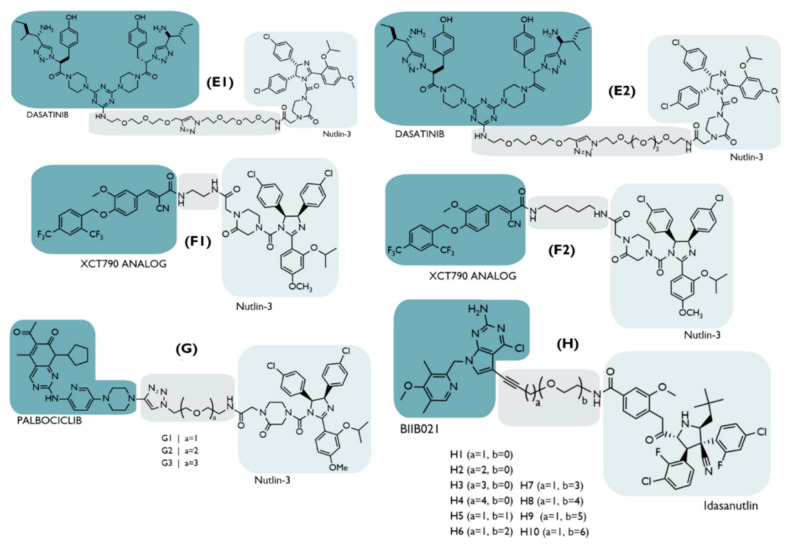
Chemical structure of some MDM2-based PROTACs intended for the treatment of breast cancer. The PROTACs E1 and E2 target the tropomyosin receptor kinase C (TrkC) [69]; the F1 and F2 PROTACs target the estrogen-related receptor α (ERRα) [70]; PROTACs G, with diverse types of linkers, degrade cyclin-dependent kinase 6 (CDK6) [71]; PROTACs H degrade heat shock protein 90 (HSP90) [72].

### 4.6. MDM2-Based PROTAC—Lung Cancer

In 2020, MDM2-based PROTACs, which target the EGFR (with mutations between leucine 858 to arginine 858 and threonine 790 to methionine 790), were created, being designated as EGFR^L858R/T790M^ [73].

The purpose of these PROTACs is to promote the degradation of this protein, which is known to be a defined therapeutic target for the treatment of non-small cell lung cancer (NSCLC) [73].

For the treatment of NSCLC, the Food and Drug Administration (FDA) has already approved a series of EGFR inhibitors, such as gefitinib, erlotinib, osimertinib, etc. [73]. However, as a result of mutations that occur in the target protein, the inhibitors are prevented from exerting their therapeutic action, such that 40% of patients with NSCLC end up developing resistance [73].

To try to counter this resistance phenomenon, the development of PROTACs, which promote the degradation of the EGFR^L858R/T790M^ target, may be an alternative. Thus, several PROTACs were synthesized, recruiting different E3 ligases with different ligands [73].

Focusing only on the PROTACs that recruit MDM2 (PROTACs I_1–5_), these use the inhibitor idasanutlin to bind to E3 ligase [73]. To recruit the EGFR^L858R/T790M^ target, the XTF-262 inhibitor was incorporated into PROTAC [73]. It should be noted that this inhibitor has enormous selectivity for this target [73]. Both these ligands are linked to each other through various types of linkers in order to find out the impact they may have on PROTAC performance (Figure 11) [73].

When evaluated in a cell line that presents wild-type EGFR, none of the five PROTACs were shown to degrade the target (DC_50_ > 2000 nM), demonstrating that the selectivity was desired by the incorporation of the inhibitor XTF-262 [73]. When tested on the H1975 cell line, which has the EGFR^L858R/T790M^ protein, compounds I1, I4, and I5 have not been shown to degrade the target (DC_50_ > 2000 nM) [73]. However, PROTACs I3 and I2, which have a larger linker than the previous ones, show DC_50_ values of 77 and 263.8 nM, respectively [73].

Although other PROTACs recruiting other E3 ligases showed better results, this showed that linker length and composition are critical aspects in compound development. This is because, when the linkers are too small, they can lead to the formation of dimers (PROTAC-POI or PROTAC-MDM2) being favored over the formation of a necessary ternary complex (POI-PROTAC-MDM2) [67]. On the other hand, very long linkers may not favor the interaction needed to promote the ubiquitination of the target protein by the E3 ligase, given the greater freedom of movement [67].

### 4.7. HOMO-MDM2-Based PROTACs

The designation of homo-PROTACs is given to all those whose target and E3 ligase is the same entity [65]. In this case, the target of a HOMO-MDM2-based PROTAC is MDM2 itself, which will be ubiquitinated and then degraded, with a consequent increase in the levels of the tumor suppressor p53 [65].

Based on the knowledge that approximately more than half of existing cancers result from p53 disorders and that MDM2 is overexpressed in many resistant cancers with a poor prognosis, its degradation could be a new therapeutic strategy [65]. At the moment, there are several MDM2 inhibitors [63]. However, they have many adverse effects, such as the risk of developing hematological disease [65]. The degradation of the MDM2 enzyme has the advantage that the necessary concentrations are relatively lower than those used for its inhibition, given the substoichiometric catalytic capacity of PROTACs, resulting in greater safety [65].

Therefore, in 2021, several PROTACs consisting of two units of a nutlin-3 derivative linked together by distinct types of ligands, were synthesized and evaluated for use in the treatment of NSCLC [65].

Of all the synthesized compounds (Figure 12), PROTACs J9 and J10a-d have alkyl chains of an increasing length [65]. However, the longer the chain, the lower the binding affinity for the targets [65]. When switching to a PEG-based linker (PROTACs J10e-f), there is an increase in binding capacity, and it is maximum when a flexible imino group is introduced (J11a: Ki = 0.1 mmol/L) [65]. By decreasing the chain (J11b-c), there is a decrease in affinity, with the compound J11a being the most potent of all [65].

When evaluated in four different tumor cell lines, the results allowed us to infer that the compounds with the highest affinity are the ones with the highest antitumor activity [65]. More specifically, those that present a PEG linker have superior in vitro activity, and J10a, J10f, and J11a appear to be more potent than the positive control [65]. When tested in lung tumor cell lines (A549), PROTAC J11a, in 24 h, induced the degradation of more than 95% of MDM2, with a concentration of 2 µmol/L, presented a DC_50_ value of 1.01 µmol/L [65]. Moreover, it has been shown to induce apoptosis in about 24% of A549 cells exposed to 5.0 µmol/L for 24 h [65].

Once the J11a compound presents enantiomers due to the chiral center in the imi-dazole group, its separation by a chiral column and subsequent evaluation, allows us to conclude that the J11a-1 enantiomer presents the best conformation to bind to the MDM2, immediately starting to degrade it after the end of the 2 h after exposure [65]. When tested in vivo, PROTAC J11a-1 was shown to inhibit tumor growth by 52% after an intraperitoneal administration of 30 mg/kg, twice a day, for 21 days [65]. In terms of p53 levels, they increased, as opposed to MDM2 levels, which decreased [65].

In sum, this is the first MDM2-based PROTAC capable of promoting the degradation of MDM2 itself, in vitro and in vivo, with great antitumor efficacy [65]. However, further studies are still needed to prove that PROTACs are safer than conventional inhibitors.

### 4.8. Peptidic-MDM2-Based PROTACs

Designated as ^PMI^BCR/Abl-R6, this is a peptide PROTAC capable of degrading Bcr/Abl, a protein of great importance in Philadelphia chromosome-positive chronic myeloid leukemia (LK-Ph+), which is an abnormal chromosome that leads to the increased expression of the Bcr/Abl fusion protein, which is related to 95% of chronic myeloid leukemia (CML) and 20% of acute lymphoblastic leukemia (ALL) cases [74,75].

The ^PMI^BCR/Abl-R6 results from the junction of PMI, a peptide inhibitor of the MDM2/p53 interaction to the α1 helix of the tetramerization domain of Bcr/Abl [74]. To increase its cell permeability, a poly-arginine cationic tail with six aa residues was added [74]. The present PROTAC has been shown to prevent the oligomerization of Bcr/Abl, which is necessary for the activation of intracellular signaling pathways that cause cell proliferation [74]. Moreover, by recruiting MDM2, this promotes the polyubiquitination of the fusion protein, inducing its degradation by the proteasome [74,75]. By occupying MDM2, it promotes an increase in p53 levels, which culminates in the activation of apoptotic pathways in the tumor cell (Figure 13) [74,75].

When compared with the use of the PMI inhibitor alone in the HCT116 cell line, it was found that PMI cannot kill cells, given its poor stability and inability to permeate the membranes, unlike PROTAC, which, in addition to being more stable, is also more permeable [74]. Currently, there are several inhibitors for Bcr/Abl, such as imatinib, dasatinib, nilotinib, among others [75]. However, the high toxicity and the development of imatinib-resistant leukemias lead to a need to create new therapeutic weapons [75].

Therefore, tests in imatinib-resistant LK-Ph+ cell lines and KU-812 (CML) and SUP-B15 (ALL), with the wild-type p53 gene and overexpression of MDM2, demonstrated that PROTAC was able to induce apoptosis, inhibiting more than 80% cell viability in both cell lines, with IC_50_ values of 19.6 µM and 8.1 µM, respectively [75]. On the other hand, imatinib and the PROTAC derivative without the PMI—named BCR/Abl-R6 (that is, without being able to recruit MDM2)—did not present significant results [75]. Additional studies have shown that ^PMI^BCR/Abl-R6 is not toxic in LK-Ph(-) cell lines, does not affect target expression, and is able to promote the degradation of different isoforms of the Bcr/Abl protein and even some mutant forms that confer imatinib resistance, which is accompanied by increases in p53 levels [74,75]. When tested in an ex vivo study with human CML and ALL cells, and within in vivo with-mice xenograft models, the PROTAC demonstrates the enormous potential for the treatment of LK-Ph+ and even in some forms that have resistance to imatinib, being non-toxic and non-immunogenic and outperforming the results obtained by either nutlin, imatinib, or BCR/Abl-R6 [74,75]. It should be noted that PROTAC has a higher molecular weight than the previous ones, and the fact that it has a peptide origin may confer some lack of stability or difficulty in membrane permeation. However, it is known that peptide drugs, in general, have better specificity and affinity, as they have a larger area for interaction and have less toxicity than synthetic drugs [74,75].

As a powerful compound with a future in the treatment of leukemia, this peptide MDM2-based PROTAC is proof of the enormous potential that the association of target degradation with the recruitment of MDM2 can offer, not just in those of a chemical origin.

## 5. Design of MDM2-Based PROTACs

Although it is unclear what may have contributed to the failure of some of the previous PROTACs in inducing the degradation of their targets, it is necessary to keep in mind that any changes made to one of the three modules of PROTAC—target ligand, linker, and the ligand of E3 ligase—can be decisive in terms of its responsiveness, being able to increase or decrease aspects such as its potency, efficacy, selectivity, stability, solubility, bio-availability, among many others [12,19,55].

As previously discussed, linker characteristics, such as length or chemical composition, play a key role in aspects such as the selectivity or effectiveness of PROTACs, as they affect the way ligands interact with their targets [12,19]. The rigidity of the linker, as well as its composition or length, are aspects that reveal the impact of not only the selectivity for POIs but also the metabolic stability or the aqueous solubility of PROTAC. With regard to rigidity, it may be essential to fix PROTAC in its bioactive conformation, increasing the degradation of POI [10,11]. Additionally, the properties of the linker, such as for example, its lipophilicity, must be adjusted to those of the ligands [19,55].

Besides the linker, ligands responsible for recruiting MDM2 and POI are relevant [12,19,55]. Preferably, these should be, among other characteristics, specific, safe, and small [19]. Additionally, they should not have high binding affinities because it makes it difficult to dissociate the target or MDM2, and this can compromise the catalytic mechanism of action of PROTAC [86]. Relative to the MDM2 ligand, it must be able to bind to the ligase without compromising its ability to ubiquitinate the target [12].

The choice of the E3 ligase for recruitment is essential, and the ideal case, from a developmental point of view, is to recruit one that is overexpressed in a specific type of cancer, accompanied by the overexpression of POI in those same cells, which is not significantly expressed in normal cells [19,64]. Following this line of thinking, a group of researchers has identified a number of cancers and promising therapeutic targets for MDM2-based PROTACs, such as the BRD4, cyclin-dependent kinase 9 (CDK9) or induced myeloid leukemia cell differentiation (MCL1) proteins in stomach cancer; phosphatidylinositol 3-kinase catalytic subunit type 3 (PIK3C3) in BC; and MCL1 in BC and lung adenocarcinoma [64]. Therefore, the creation of new MDM2-based PROTACs to degrade the previously mentioned targets could be of great interest in terms of cancer treatment.

In summary, although there is no rationalized design regarding the development of MDM2-based PROTACs, we can say that their empirical development is very dependent upon the choice of ligands selected to recruit both MDM2 and POI and the linker typology chosen for their junction. In addition, it must be taken into account that the choice of li-gands that have good affinity with the target is not always enough to guarantee the success of PROTACs, which is defined by obtaining the lowest possible DC_50_ value, with concentrations of PROTAC preferably in the order of nanomolar to picomolar and with the greatest possible degradation [10].

## 6. Advantages of MDM2-Based PROTACs

Nowadays, conventional therapies with mAbs, capable of blocking extracellular targets or SMIs which can inhibit extra and intracellular compounds (and that respect Lipinski’s Rule of 5 (Ro5) for oral administration), unfortunately have some limitations. [87].

However, SMIs perform a stoichiometric inhibition [22], in which one molecule of inhibitor inhibits one molecule of the target, following an occupancy-driven model, in which high concentrations are required to maintain a certain level of occupancy of the target to exert the desired effect [19,22]. Consequently, the use of high concentrations promotes the occurrence of adverse effects associated with some disadvantages, such as the inability to inhibit “undruggable proteins”, protein accumulation, and increasing its expression or the development of resistance [19,87].

The development of PROTACs, more specifically MDM2-based PROTACs, can promote therapeutic effects through target degradation, as seen above, and has some advantages over inhibition, such as the following.
PROTACs follow the event-driven model—a molecule of PROTAC can degrade several molecules of POI. Thus, this presents a catalytic cycle of degradation, allowing for its use with substoichiometric concentrations [65,86];PROTACs can display cooperativity—binding to a second protein in PROTACs can lead to the formation of the ternary complex being favored (positive cooperativity), or it can lead to the formation of binary complexes (negative cooperativity) or simply remain not affected (neutral cooperativity). If positive cooperativity exists, a stable ternary complex can be formed even in the presence of low affinities with the POI [19,22];Fast, selective, and sustained POI degradation—selective degradation, even for the isoforms of the target that may not have an active site, (with the use of low concentrations (µM—nM)) is associated with a reduction in the activation of the respective signaling pathways [65,67,74,75];Ability to cause cellular apoptosis, delay the cell cycle, decrease proliferation and metastasis of tumor cells—the activation of the p53 suppressor pathway in vitro, ex vivo, and in vivo [65,67,74,75];PROTACs do not cause a compensatory expression of POI, not even intracellular accumulation—there are no changes in the mRNA expression levels of the respective target protein [67,74,75];PROTACs are less susceptible to the development of resistance—the resistance me-chanism will have to encompass both of the pathways of the PROTAC, given that, in the absence of the degradation capacity, the inhibition of the target or MDM2 by itself may still have anticancer action [67].

## 7. Disadvantages of MDM2-Based PROTACs

For a PROTAC to be able to exert its action, it must go inside the cell, and this can be a limiting step given that they are usually large molecules [86]. As seen above, the strategy of reducing the size of the ligands or adding peptides that promote membrane permeability (poly-arginine tail) can help to overcome these difficulties [86].

Furthermore, PROTACs, when in high concentrations, can develop the hook effect [67]. However, if this presents positive cooperativity, then this effect can be attenuated, favoring the formation of the ternary complex and reducing the required concentrations [19].

From a pharmacokinetic viewpoint, PROTACs are usually given by parenteral administration [87]. Although not following Lipinski’s Ro5, given that MDM2-based PROTACs have the highest molecular weight and most lipophilic type of PROTACs, with 3 aromatic rings in the MDM2 ligand, their oral administration will be a challenge, but, currently, there are examples of PROTACs that may have good bioavailability when administered orally, and MDM2-based PROTACs may become good candidates for oral anticancer therapies [55]. In addition to absorption issues, its metabolism is also an aspect to be taken into account since, in vivo, they will give rise to metabolites, which, even if they are not able to degrade the target, may perhaps inhibit POI or MDM2 [55]. Furthermore, the fact that these PROTACs have a high molecular weight may hinder their ability to enter the interior of liver cells and, thus, be metabolized; in this sense, studies to assess this ability and predict its impact are necessary [14].

Another relevant issue is related to the fact that genotoxic stress promotes the expression of MDM2 isoforms that lack the full N-terminal p53 binding domain and varying extensions of the central acid domain, resulting from alternative splicing, with the consequential loss of its ubiquitinating activity [61]. However, as far as we know, there are no studies that allow us to understand the impact that these isoforms may have on the performance of MDM2-based PROTACs [61].

Therefore, MDM2-based PROTACs offer a full set of aspects that can be improved, making it necessary to conduct more studies to clarify, create, and improve these promising molecules in therapy.

## 8. Future Perspectives and Challenges

Although there are a small number of MDM2-based PROTACs, this type of PROTAC represents a whole new universe to explore, given the advantages that recruiting MDM2 presents.

Therefore, it is expected that in the future, new compounds will be designed to degrade new proteins relevant to cancer development, as well as exploring the possibility of creating PROTACs capable of being controlled in space and time, increasing their specificity and safety, with the incorporation of MDM2 ligands in phosphoPROTACs, optoPROTACs, PHOTACs, or even conjugating them with antibodies.

To overcome the problems related to its large dimensions, MDM2-based CLIPTACs can also be created. This type of molecule can form an MDM2-based PROTAC in situ.

Although uncertain, the future of MDM2-based PROTACs is very promising.

## 9. Conclusions

The MDM2-based PROTACs, although still under-represented within the universe of PROTACs, have unique and very advantageous anti-tumor characteristics, given that, in addition to degrading proteins, such as the AR [20], BCR4 [67], PARP1 [68], or their own MDM2 [65], they have the ability to activate the p53 tumor suppressor pathway. This leads to the consequent activation of apoptosis processes in tumor cells, leading to their cell death, presenting, in the end, a synergistic effect—an effect much superior to the use of isolated inhibitors [67]. Therefore, with a dual mechanism of action, which requires the formation of a stable ternary complex, MDM2-based PROTACs, with some refinement, may well become candidates for the future anticancer therapeutic arsenal, as they have already proven in certain cases to be able to be effective in vitro and in vivo at concentrations in the nanomolar order, given their catalytic and synergistic mode of action, and with characteristics that overcome the current disadvantages of conventional therapies, highlighting the lack of selectivity for the target and cell type, the development of resistance, tolerance, accumulation of the target protein, and processes of up-regulation of the target [65,67,74,75].

Furthermore, the great advantage of PROTACs, and of those that recruit MDM2, lies in the possibility of degrading previously “undruggable proteins”, that is, they allow for the degradation of targets which do not have an active site, thus being able, with these new molecules, to interfere with new signaling pathways crucial for the development and proliferation of cancer cells, something that has never been explored before [9].

Therefore, the MDM2-based PROTACs need more studies to be improved, present new characteristics, as well as overcoming their current limitations. However, everything points to these molecules having an incredible potential to become an innovative therapeutic strategy for cancer treatment.

## Figures and Tables

**Figure 1 ijms-23-11068-f001:**
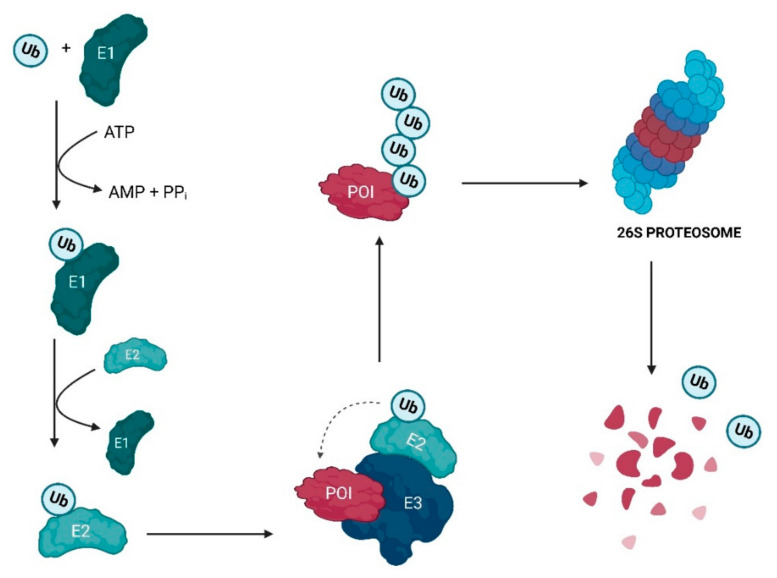
The ubiquitination process by the Ubiquitin-Proteasome System (UPS)—the entire ubiquitination process begins with the activation of the ubiquitin protein (Ub) which, by covalently binding to the ubiquitin-activating E1 enzyme, is activated in an ATP-dependent process. Upon activation, Ub is transferred to the ubiquitin-conjugating enzyme E2. Then, the simultaneous binding between E3 ubiquitin-ligase, with the protein of interest (POI), and E2-Ub occurs, with the consequent ubi-quitination of POI. The POI, being poly-ubiquitinated, is thus recognized by the 26S proteasome, resulting in its degradation [19,21,22,24].

**Figure 2 ijms-23-11068-f002:**
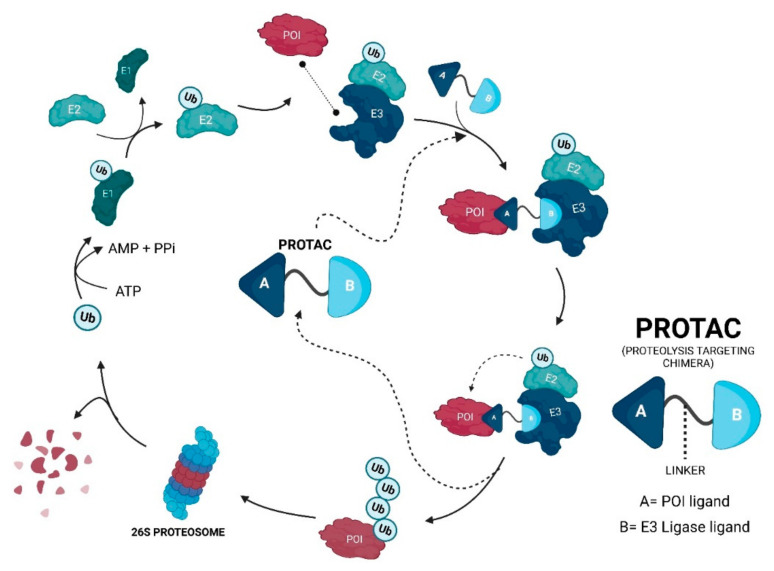
The PROTACs mechanism of action—PROTAC promotes the simultaneous and artificial binding between the protein of interest (POI) and the chosen E3 ligase, forming a stable ternary complex. This promotes the ubiquitination of the POI and the subsequent degradation by the UPS. After the degradation of a target molecule, PROTAC has the ability to re-link with other POI molecules in case it has ligands that reversibly bind to the target and restart a whole new cycle of degradation [10,12,19,27].

**Figure 4 ijms-23-11068-f004:**
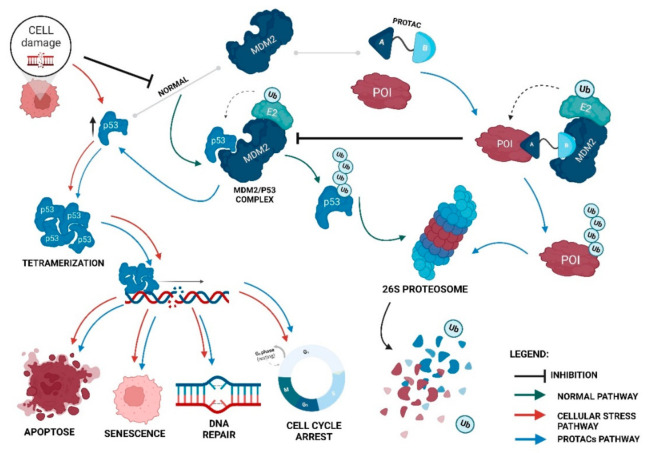
The impact of MDM2-based PROTACs on the MDM2/p53 cycle. Under normal conditions, p53 levels are kept low due to the presence of the MDM2 E3 ligase, which, when binding to the tumor suppressor, promotes its polyubiquitination, being subsequently recognized by the 26S proteasome, and degraded. When cell damage occurs, p53 is stabilized and forms a tetramer, and its ubiquitination by MDM2 is prevented. The tetramer is transported towards the nucleus, where it activates the transcription of those genes responsible for the processes of apoptosis, senescence, DNA repair, and cell-cycle arrest. However, in the presence of an MDM2-based PROTAC, when recruiting MDM2, this prevents it from binding to its natural substrate, p53, increasing the levels of the suppressor, which culminates not only in the degradation of POI, but also in the activation of processes that prevent the development of cancer cells [8,12].

**Figure 5 ijms-23-11068-f005:**
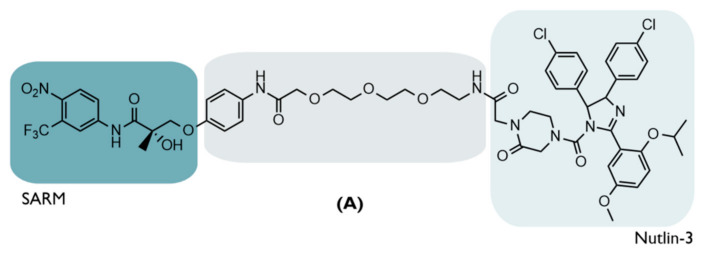
Chemical structure of PROTAC A. The first MDM2-based PROTAC recruits its target, the androgen receptor (AR), through a selective AR modulator (SARM), hydroxyflutamide. This SARM is linked to nutlin-3, which is responsible for recruiting the MDM2, through a linker derived from PEG, with 13 atoms [20].

**Figure 6 ijms-23-11068-f006:**
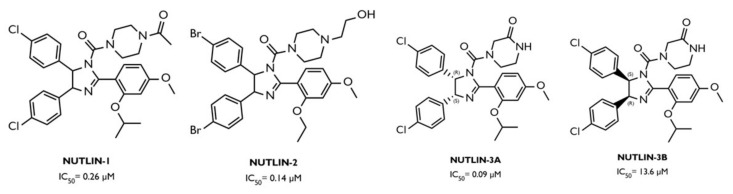
Nutlins. Chemical structures of the different compounds of the nutlins family, with the respective IC_50_ values, resulting from the inhibition of MDM2 [63].

**Figure 7 ijms-23-11068-f007:**
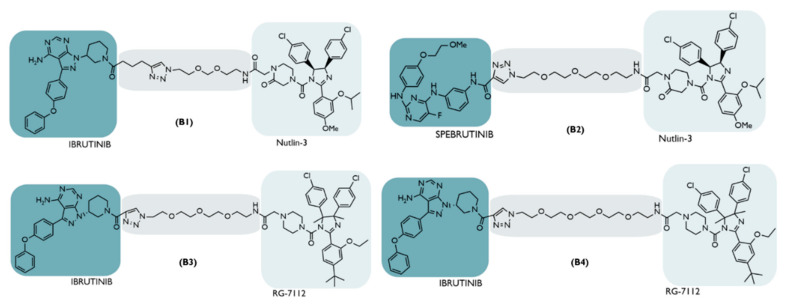
Chemical structures of PROTAC B. Recruiting Bruton’s tyrosine kinase (BTK) via the inhibitor ibrutinib (B1,B3,B4) or spebrutinib (B2); PROTAC B degrades the target, recruiting MDM2 via nutlin-3 (B1,B2) or the inhibitor RG-7112 (B3,B4) [66].

**Figure 8 ijms-23-11068-f008:**
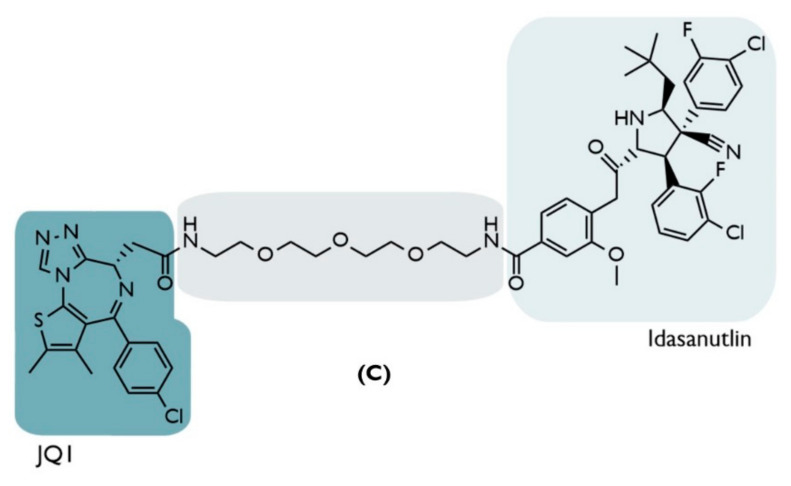
Chemical structure of PROTAC C. Recruiting Bromodomain Containing 4 (BRD4) via the JQ1 inhibitor, PROTAC **C** degrades the target, recruiting MDM2 via the inhibitor idasanutlin. Both inhibitors are linked together through a 13-atom PEG derivative [67].

**Figure 9 ijms-23-11068-f009:**
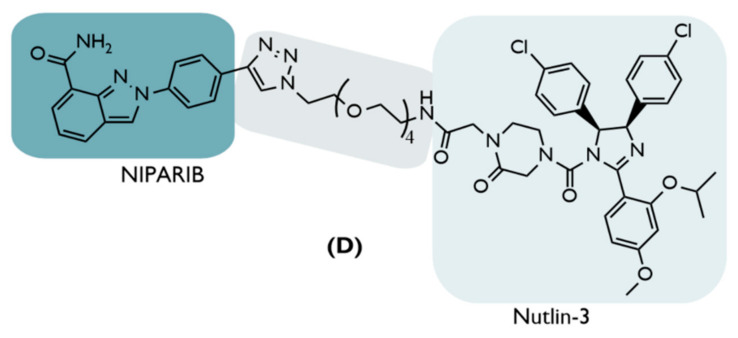
Chemical structure of PROTAC D. Recruiting Poly (ADP-ribose) polymerase-1 (PARP1) through the niparib inhibitor, PROTAC D, degrades the target, recruiting MDM2 via the inhibitor nultin-3. Both inhibitors are linked together through a PEG derivative [68].

**Figure 11 ijms-23-11068-f011:**
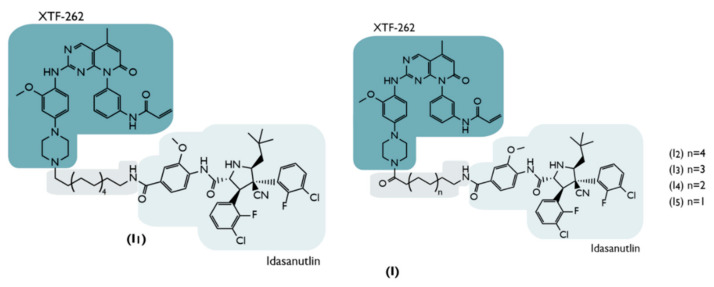
Chemical Structures of PROTACs I. Recruiting the mutated epidermal growth factor receptor protein (EGFR^L858R/T790M^) through the XTF-262 inhibitor, and E3 ligase MDM2 through the idasanutlin inhibitor; PROTAC I has different types of linkers that allow us to assess the impact that these have at the level of the biological response generated by the compound [73].

**Figure 12 ijms-23-11068-f012:**
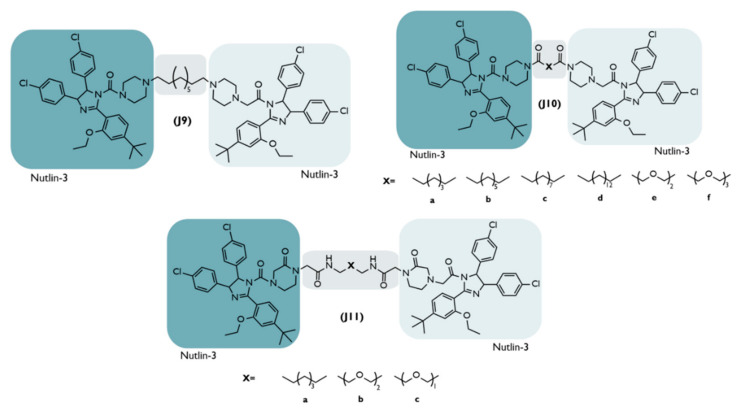
Chemical structure of HOMO-MDM2-based PROTACs. PROTACs J9, J10, and J11 have the same ligands at both ends, which allows them to recruit MDM2 and promote their own degradation [65].

**Figure 13 ijms-23-11068-f013:**
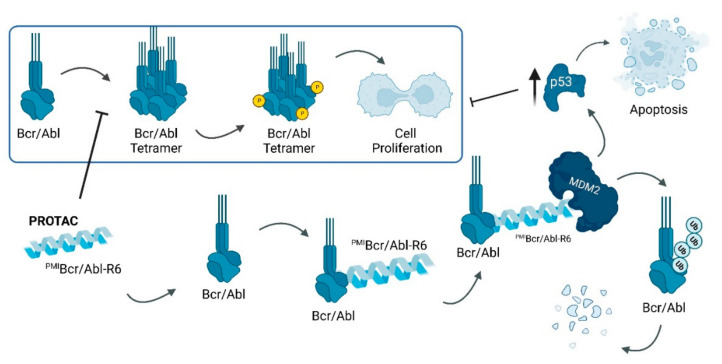
Mechanism of action of PROTAC ^PMI^BCR/Abl-R6. In a normal situation, the Bcr/Abl protein tetramerizes and then phosphorylates, resulting in the activation of the signaling pathways that promote cell proliferation. In the presence of PROTAC, the Bcr/Abl protein is degraded and, therefore, does not tetramerize, preventing the activation of cell proliferation. On the other hand, the increase in tumor suppressor p53 promotes cellular apoptosis [74,75].

**Table 1 ijms-23-11068-t001:** The role of p53 in different cellular biological processes.

Process	Mechanism	[Ref]
**Damaged DNA**	Genes, such as *PUMA* and *BTG2*, responsible for DNA repair, apoptosis, cell-cycle arrest and senescence, are transcribed with p53 activation.	[56]
**Metabolism**	Metabolic reprogramming is prevented by p53.	[57]
**Autophagy**	The p53, through the activation of genes such as *ATG10*, can promote the degradation of damaged cell organelles, inhibiting the process of tumor formation.	[57]

**Table 2 ijms-23-11068-t002:** The different domains of the MDM2 protein.

Domain	Mechanism	[Ref]
**N-terminal p53** **Binding domain**	Interacts with p53.	[58]
**Nuclear localization** **signal**	Transport of MDM2 from the cytoplasm to the nucleus.	[58]
**Nuclear export** **signal**	Transport of MDM2 from the nucleus to the cytoplasm.	[58]
**Acidic domain**	It induces p53 degradation by being phosphorylated.	[58]
**Zinc-finger** **domain**	Regulation of p53 levels: interacts with ribosome proteins that bind to the acidic domain and inhibit p53 degradation; it induces suppressor ubiquitination after MDM2-p53 binding.	[58]
**C-terminal RING-Finger** **domain**	It induces suppressor ubiquitination after MDM2-p53 binding.	[58]

## Data Availability

Not applicable.

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
