# Peer review of "MDM2-Based Proteolysis-Targeting Chimeras (PROTACs): An Innovative Drug Strategy for Cancer Treatment"

_ijms, 2022, doi:10.3390/ijms231911068_

Round 1

Reviewer 1 Report

The manuscript entitled MDM2 based Proteolysis-Targeting Chimeras (PROTACs): an Innovative Drug Strategy for Cancer Treatment” analyze and discuss the characteristics of MDM2 based PROTACs developed for the degradation of oncogenic proteins. However, there are some issues that need to be addressed to make this review more complete. 

1.     There are many abbreviated forms along the manuscript. I suggest that a list of abbreviations should be provided in front of the article.

2.     Some recent reviews should be added, such as Chem. Soc. Rev. 2022, 51, 5214-5236; Nat. Rev. Drug Discov. 2022, 21, 181-200; Chem. Soc. Rev. 2022, 51, 7066-7114 and Future Med. Chem. 2020, 12, 915-938.

Author Response

Dear Reviewer #1

We would like to thank the reviewer #1 for his review and comments, which were most helpful in improving our manuscript. All the Reviewer #1 comments were followed, and we are now submitting our reply.

Reviewer #1 note:

  1. There are many abbreviated forms along the manuscript. I suggest that a list of abbreviations should be provided in front of the article.

               R: The reviewer can find the list of abbreviations added at the end of the article (line 1000).

  1. Some recent reviews should be added, such as Chem. Soc. Rev. 2022, 51, 5214-5236; Nat. Rev. Drug Discov. 2022, 21, 181-200; Chem. Soc. Rev. 2022, 51, 7066-7114 and Future Med. Chem. 2020, 12, 915-938.

R: All suggested references have been properly referenced throughout the text of this article and are present in the updated reference list (REF: 10,11,14,15).

Reviewer 2 Report

The review article entitled "MDM2 based proteolysis-Targeting Chimeras (PROTACs): an Innovative Drug Strategy for Cancer Treatment." Investigation of this review will provide some insights into cancer treatment.  However, the current version should be revised. The comments are as follows:

1. Author could write more in depth in introduction section

2. Line 50: RNAi (RNA interference)

3. Figure 1 is not clear, normal audience cannot understand (bit confusing) so, author could  write more details in Figure 1- legend or represent figure very clear.

4. Evolutionary perspective of PROTACs, Can author illustrate diagrammatically so that researchers can understand easily. 

5. Author could discuss more details about  PROTACs molecules such as stability, degradation, solubility, concentration and ADME properties.

6. Author could explain complex structure of MDM2 and P53? also, write more details in interaction between ligand and protein molecules.

7. Author could explain MDM2 domains and briefly describe other domains of MDM2 and P53 fold in relations to each other?

8. Author could describe more details in both in vitro and in vivo data of how MDM2 promote/prevent tumor development?

9. Author could explain how MDM2 evolve both advantage/disadvantage?

10. Author could describe how does stress influence MDM2 expression?

11. how to understand the degradation activity and how to rationally design PROTACS?

12. Author could explain any novel new idea for cancer treatment.

Author Response

Dear Reviewer,

The authors would like to thank reviewer 2 very much for reviewing our manuscript and considering it for resubmission.

We analyzed and answered your questions and comments. Please find those answers in the reports attached below.

Reviewer 3 Report

The review is a high quality work, important for experts and understandable for a broad audience. The authors analyzed a vast set of literature and provided very clear-cut and important new insights into the problem. 

Acceptance is recommended. 

Author Response

Dear Reviewer #3

We would like to say that we are very grateful for your review and comments.

Round 2

Reviewer 2 Report

Thank you for all review comments and answers and highly appreciate for authors efforts.